



# Mixing state, spatial distribution, sources and photochemical enhancement to sulfate formation of black carbon particles in the Arctic Ocean during summer

Longquan Wang[1,2], Jinpei Yan[3], Afeng Chen[1,4], Bei Jiang[1,5], Fange Yue[1], Xiawei Yu[1], Zhouqing Xie[1,6*]

[1]Anhui Key Laboratory of Polar Environment and Global Change, Department of Environmental Science and Engineering, University of Science and Technology of China, Hefei 230026, China.

[2]Department of Carbon Neutral Science and Engineering, Anhui University of Science and Technology, Hefei 230026, China.

[3]Third Institute of Oceanography, Ministry of Natural Resources, Xiamen 361005, China.

[4]Engineering and Technological Research Centre of National Disaster Relief Equipment, Army Logistics Academy, Chongqing, 401331, China.

[5]College of Ecology and Environment, Xinjiang University, Urumqi, 830017, PR China.

[6]State Key Laboratory of Fire Science, University of Science and Technology of China, Hefei, 230026, China.

*Correspondence to*: Zhouqing Xie (zqxie@ustc.edu.cn)



**Abstract.** Black carbon heats the atmosphere by absorbing solar radiation and regulates the radiation balance of the Earth. Specifically in the Arctic region, black carbon accelerates Arctic warming by simultaneously altering surface albedo. Nonetheless, assessing the climatic impacts of black carbon aerosols in the Arctic is challenging due to their considerable variability in temporal and spatial distribution, sources, and chemical composition. Black carbon particles (0.2-2μm) in the Arctic Ocean were investigated using a ship-based single particle aerosol mass spectrometer from July to August 2017. In the central Arctic Ocean, near the Norwegian Sea-Iceland and the North Atlantic, biomass combustion is the predominant source of black carbon particles, constituting over 50%, with a particularly high contribution exceeding 70% in the central Arctic Ocean. Within the Chukchi Sea region, terrestrial transport from mid and low latitudes emerges as the primary source of black carbon particles, representing over 50%, with biomass combustion and anthropogenic pollution sources each contributing around 25%. Near Svalbard, biomass combustion sources and terrigenous transport stand out as the primary sources of black carbon particles, with their contributions being comparable. Furthermore, the ratio of sulfate to nitrate in black carbon particles was notably higher compared to that in sea salt particles. This ratio increased with elevated black carbon content and sunlight intensity, suggesting that Arctic black carbon particles substantially facilitated sulfate formation through photochemical processes. Such interactions could potentially modify the mixing state of Arctic black carbon particles and their radiative impacts.

## 1  Introduction

Black carbon, an amorphous form of carbon produced by the incomplete combustion of carbon-containing fuels or biomass, plays a crucial role in atmospheric physical chemistry by absorbing solar radiation and influencing Earth's radiation balance. Additionally, the warming effect of black carbon modifies the vertical stability of the atmosphere and alters cloud distribution, which further impacts the radiation balance (Ramanathan & Carmichael, 2008). The deposition of black carbon on ice and snow surfaces significantly reduces their albedo, accelerating the melting process. This is particularly pronounced in the Arctic region, where black carbon contributes to warming not only by heating the atmosphere but also by changing the surface albedo and through other direct and indirect radiative effects (Flanner et al., 2007; Serreze & Barry, 2011). Arctic black carbon aerosol, characterized as a transportable



48 species with various potential sources, exhibits distinct spatial and seasonal distribution patterns (Matsui

49 et al., 2022). These variations are critical for assessing its impact on radiative forcing. Furthermore, the

50 mixing state of particles containing black carbon significantly influences their radiative effects (Jacobson,

51 2001), making detailed observations and analyses essential.

52   The mass concentration of black carbon aerosols in the Arctic near-surface atmosphere exhibits

53 significant seasonal variability, typically higher in winter and spring, and lower in summer and autumn

54 (Qi et al., 2017; Sharma et al., 2006; Sharma et al., 2004; Sharma et al., 2019). Influenced by atmospheric

55 transport patterns, the sources and vertical structures of black carbon aerosols also vary seasonally.

56 During winter and spring, the Arctic near-surface atmosphere is predominantly affected by transport from

57 higher latitudes, facilitating the movement of black carbon from Siberia and Europe through low-altitude

58 pathways to the bottom of the Arctic troposphere. Conversely, black carbon from mid-latitudes reaches

59 the middle and upper troposphere of the Arctic via long-distance transport. In summer, increased

60 precipitation in middle and high latitudes, which impedes long-distance transport, shifts the primary

61 source of atmospheric black carbon to biomass burning within the Arctic and neighboring regions such

62 as Siberia and Alaska (Evangeliou et al., 2016; Matsui et al., 2022; Wang et al., 2014). Although

63 atmospheric concentrations of black carbon are lower in summer than in winter, the higher frequency of

64 summer precipitation leads to quicker removal rates and higher deposition fluxes of atmospheric black

65 carbon (Willis et al., 2018). Observational data on atmospheric black carbon aerosol concentrations in

66 the Arctic also demonstrate notable spatiotemporal variations. Long-term continuous measurements at

67 Barrow and Alert stations in the Arctic revealed that the average concentration of black carbon aerosols

68 from 1980 to 2003 was approximately 25 ng/m³ (Sharma et al., 2004). Meanwhile, the average annual

69 concentration measured at Zeppelin Station from 1998 to 2007 was 39 ng/m³ (Eleftheriadis et al., 2009),

70 significantly higher than the former. Additionally, the spatial distribution of black carbon in Arctic ice

71 and snow varies, with concentrations decreasing from the Russian Arctic, through the Canadian and

72 Alaskan Arctic, to the Arctic Ocean and Greenland, highlighting a significant decline from the Arctic

73 coastal regions to the central Arctic Ocean (Dou & Xiao, 2016). These temporal and spatial discrepancies

74 in Arctic black carbon aerosols underline the importance of long-term, multi-regional observations in the

75 Arctic.

76   Evaluating the climatic impact of black carbon aerosols in the atmosphere and on snow cover is



crucial for understanding Arctic climate change. Unlike greenhouse gases such as carbon dioxide and
methane, black carbon aerosols possess a broader absorption spectrum, capable of absorbing radiation
across infrared to ultraviolet wavelengths (Bond & Bergstrom, 2006). In the IPCC 5th Assessment Report,
the direct radiative forcing of black carbon aerosols was estimated at 0.4 W/m^2, with a range of 0.05 to
0.8 W/m^2. The effective radiative forcing, which includes direct radiative impacts along with climate
feedbacks and rapid adjustments, was initially estimated at -0.45 ± 0.5 W/m^2. This estimate has since
been revised to 0.063 W/m^2, ranging from -0.28 to 0.42 W/m^2, reflecting a deeper understanding of
black carbon's effects on clouds and water vapor. Research indicates that the direct radiative forcing from
black carbon aerosols originating from fossil fuels is comparable to 78% of that from carbon dioxide,
while those from biomass burning equate to about 58% of carbon dioxide's effect (Hansen et al., 2005).
Considering indirect effects like changes in ice and snow albedo and cloud properties, the black and
organic carbon components from fossil fuel soot exert a net positive global radiative forcing. In contrast,
those from biomass combustion have a net negative global radiative forcing (Hansen et al., 2005). In the
Arctic, black carbon aerosols starkly contrast with the bright ice and snow surfaces, enhancing their solar
radiation absorption and leading to higher direct radiative forcing. Studies suggest that in the Arctic, the
warming effect of black carbon aerosols surpasses the cooling effect of reflective aerosols, making them
a potent warming agent in the region (Quinn et al., 2007). Beyond atmospheric warming, black carbon
in snow can intensify climate change by modifying the albedo feedback of snow and ice. The deposition
of black carbon on snow surfaces not only increases heat absorption but also, due to near-surface
temperature inversions that restrict energy dispersal in the Arctic, this absorbed heat further warms the
near-surface atmosphere. This process accelerates ice and snow melt and alters surface albedo, thereby
amplifying Arctic warming through the ice-albedo feedback mechanism, a significant factor in Arctic
amplification (Clarke & Noone, 1985; Dou & Xiao, 2016). The estimated annual mean radiative forcing
of black carbon aerosols from various sources on Arctic ice and snow is approximately 0.17 W/m^2 (Dou
& Xiao, 2016).

In the Arctic region, the mixing state of black carbon aerosols from different sources significantly

influences their radiative effects (Jacobson, 2001; Matsui et al., 2022). The internal mixing of black
carbon aerosols with scattering aerosols such as sulfate and organic carbon can substantially alter the
optical properties of black carbon, increasing its positive radiative forcing (Chung & Seinfeld, 2002). It



has been observed that anthropogenic black carbon aerosols from Asia generally have a thicker coating compared to those from Europe, Siberia, and North America. This thicker coating is attributed to the rapid aging of Asian aerosols upon emission, often accompanied by condensable gases. Moreover, Asian anthropogenic black carbon typically travels at higher altitudes and over longer distances, resulting in a longer residence time and more thorough aging process (Matsui et al., 2022). Black carbon aerosols of different origins within the same region also exhibit variations in their mixing state. For instance, black carbon aerosols from biomass combustion in Siberia and North America tend to have a larger proportion and thicker coatings compared to those from anthropogenic sources in these areas. This difference is largely due to the transport of biomass combustion-derived black carbon to the Arctic occurring primarily in summer, which facilitates a faster aging rate than in winter (Matsui et al., 2022). Anthropogenic black carbon from Asia and biomass-burning-derived black carbon aerosols from Siberia and North America exhibit stronger positive radiative effects (Matsui et al., 2022). Additionally, the mixing state of black carbon aerosols affects nucleation and wet clearance processes, thereby indirectly influencing their radiative effects (Ching et al., 2012; Ching et al., 2018). These findings highlight that the source and mixing state of Arctic black carbon aerosols represent key uncertainties in assessing their radiative impacts, and that field observations are crucial for accurately evaluating their climate effects.

The condensation of secondary species like sulfate, nitrate, and organic matter on the surfaces of black carbon particles significantly alters their mixing state (Ault et al., 2010). Despite this, few studies have explored the secondary processes occurring on black carbon aerosol surfaces. Black carbon is known to actively participate in some of these secondary processes. Recent laboratory research indicates that black carbon may catalyze the formation of sulfate, significantly contributing to the growth of secondary inorganic components in urban haze in China (Zhang et al., 2020). Research has also demonstrated that black carbon aerosols are photoactive, capable of releasing reactive oxygen species, including excited oxygen molecules and hydroxyl radicals, into the atmosphere. These species potentially facilitate the formation of sulfate and organic matter (Gehling & Dellinger, 2013; Li et al., 2019). Further studies have confirmed that black carbon aerosols engage in photochemical processes that enhance sulfate generation in urban settings (Zhang et al., 2021). However, such studies have not yet been extended to the Arctic region. Given the Arctic's sensitivity to climate changes influenced by aerosol species like black carbon and sulfate, this potential photochemical process could introduce significant





uncertainties in assessing the local aerosol physicochemical properties and their radiative effects.

**2    Materials and Methods**
**2.1 Research region and instruments**
Observations were conducted aboard the R/V "Xuelong" during the 8th Chinese Arctic Expedition
Research Cruise, which traversed a significant portion of the Arctic Ocean, spanning from 56.2° to
84.6°N latitude and 169.4° to 46.9°W longitude. The cruise took place from July 30 to August 27, 2017,
and was segmented into five distinct phases as described in Wang et al. (2022) for detailed analysis based
on the ship's location, as depicted in Figure 1, and the specific sampling times and locations for each
phase are detailed in Table S1 and summarized as follows: Chukchi Sea section, denoted as Leg I, spans
from 22:00 on July 30th, 2017, to 19:00 on August 1st, 2017, covering latitudes from 66.0°N to 74.8°N
and longitudes from 169.4°W to 159.2°W. The high Arctic section, marked as Leg II, extends from 20:00
on August 10th to 5:00 on August 12th, 2017, ranging from 83.7°N to 84.6°N and from 132.0°E to
110.4°E. Svalbard Islands section, identified as Leg III, is from 2:00 on August 17th to 11:00 on August
19th, 2017, with latitudes from 82.6°N to 74.3°N and longitudes from 25.4°E to 2.1°E. Norwegian Sea
and Iceland section, referred to as Leg IV, occurs from 14:00 on August 23rd to 14:00 on August 25th,
2017, spanning from 67.0°N to 61.2°N and from 2.1°W to 25.8°W. Atlantic Ocean section, labeled as
Leg V, ranging from 14:00 on August 25th to 17:00 on August 27th, 2017, covers from 61.1°N to 56.2°N
and from 25.9°W to 46.9°W. The methodology for particle detection utilized the onboard Single Particle
Aerosol Mass Spectrometer (SPAMS), consistent with the techniques described by Li et al. (2011). Prior
to analysis, sampled particles were dried using a Nafion dryer to remove moisture. A $PM_{2.5}$ collector was
employed to filter out particles larger than 2.5 μm. The fine particles were then drawn into the vacuum
system through a critical orifice, accelerated, and focused to form a narrow particle beam. These particles
were subsequently exposed to two continuous Nd:YAG diode lasers (532 nm) to determine their
aerodynamic diameter based on their velocity. Each particle was then ionized by an Nd:YAG laser (266
nm) to generate positive and negative fragment ions. The ionization laser maintained a power density of
$1.55 \times 10^8$ W/cm^2. The resulting fragment ions were detected using a bipolar time-of-flight mass
spectrometer. For calibration of the SPAMS, polystyrene latex spheres (PSL Nanosphere Size Standards,
Duke Scientific Corp., Palo Alto) with diameters ranging from 0.2 to 2.0 μm were used. Additionally,



PbNO₃ particles with a diameter of 0.35 μm, generated by an aerosol generation and monitoring system
(AGM-1500, MSP Corporation, USA), were utilized for mass spectrum calibration. The sampling inlet,
connected to the monitoring instruments, was positioned on a mast 20 meters above the sea surface at
the bow of the vessel to minimize the influence of ship emissions. Sampling was conducted only while
the ship was in motion. A specific peak threshold was set at five units, allowing the instrument to record
a peak signal that surpassed this threshold, thus distinguishing it from background noise (<1 unit) in the
mass spectra (Zhang et al., 2021). The same instrument and samples were successfully used by Wang et
al. (2022) to investigate the mixing state, distribution, and photochemistry of iodine-containing particles
through specific identification conditions. In this study, we adopt a new perspective, focusing on black
carbon particles from the same samples, examining their sources, distribution, and relationships to sulfate
formation under distinct identification conditions, as described in detail in Section 2.2.
**2.2  Data processing of SPAMS data**
Particle size and mass spectra were analyzed using the YAADA software toolkit
(http://www.yaada.org/) and MATLAB (http://www.mathworks.com). Throughout the entire cruise, over
2,000,000 particles were sampled and sized using two continuous Nd:YAG diode lasers (532 nm). Nearly
290,000 particles were ionized using an Nd:YAG laser (266 nm), generating both positive and negative
ion mass spectrometry data during the cruise. Black carbon particles are known to produce a series of
characteristic signal peaks at integer multiples of 12 for the mass-to-charge ratio (m/z = ±12, ±24, ±36,
±48, ±60, etc.) in the spectra (Kollner et al., 2021; Qin & Prather, 2006; Roth et al., 2016; Schmidt et al.,
2017; Sierau et al., 2014). Therefore, the presence of several of these signal peaks concurrently in the
anionic or cationic mass spectrum of a particle suggests the presence of black carbon. In this study, we
established specific criteria to identify black carbon particles among all ionized particles (Beddows et al.,
2004). A particle is determined to contain black carbon if it meets one of the following conditions:
(1) The signal peaks of m/z = 12, 24, and 36 appear simultaneously in the cationic mass spectrum.
(2) The signal peaks of m/z = -24, -36, and -48 appear simultaneously in the anionic mass spectrum.
Using the specified method, approximately 80,000 particles were identified as containing black
carbon. The average spectrum of black carbon particles is displayed in Figure S1. The figure reveals that,
in addition to the peak for carbon signals, there are prominent peaks for $^{39}K^+$ and $^{40}Ca^+$ in the cation
spectrum, as well as $^{26}CN^-/^{42}CNO^-$ and $^{97}HSO_4^-$ in the anion spectrum. To categorize these particles into



distinct groups based on the presence and intensity of ion peaks in their mass spectra, we employed the
Adaptive Resonance Theory neural network algorithm (ART-2a) (Song et al., 1999). The parameters set
for ART-2a included a vigilance factor of 0.6, a learning rate of 0.05, and a limit of 20 iterations. This
clustering approach is designed to include more than 95% of the total particles when used for
classification purposes. For the descriptive statistical analysis of the data, SPSS version 20 (IBM Inc.,
USA) was utilized.
**2.3 Satellite data acquisition**
In this study, the intensity of solar radiation was derived using the surface incoming shortwave flux
(SWGDN, in watts per square meter, W/m^2) with a 1-hour time resolution. This data was sourced from
the NASA Goddard Earth Sciences Data and Information Services Center (https://disc.gsfc.nasa.gov/)
for the specific times and locations relevant to our research.

**3   Results and Discussion**
**3.1 Mixing state of black carbon particles**
Numerous studies have established a clear link between the mixing state of black carbon particles
and their radiative impacts in the Arctic atmosphere. Black carbon aerosols, originating from various
sources, each distinctly influence their mixing states and consequently their effects on radiation. (Chung
& Seinfeld, 2002; Matsui et al., 2022). Utilizing the ART-2a algorithm, black carbon particles were
classified based on the position and intensity of characteristic peaks in their mass spectra. These particles
were categorized into seven main types across the entire route, as detailed in Figure 2, which includes
the average spectra for each type:
(1) **Potassium-Cyanide (K-CN):** Displayed in Figure 2a, these particles show a dominant $^{39}K^+$

signal in the cationic mass spectrum, with a relative peak area exceeding 0.8. Although

carbonaceous cations ($^{12}C^+$, $^{24}C_2^+$, $^{36}C_3^+$) are less pronounced, significant cyanide peaks ($^{42}CN^-$

and $^{46}CNO^-$) along with inorganic and organic carbon peaks are evident in the anionic mass

spectrum. Accounting for 51.8% of all black carbon particles, these strong signals suggest a

biomass combustion origin (Silva et al., 1999).

(2) **Calcium-Nitrate (Ca-NO₃):** Figure 2b shows that these particles primarily feature a strong

$^{40}Ca^+$ signal in the cationic mass spectrum, with a significant nitrate signal ($^{46}NO_2^-$) and



additional carbon peaks in the anionic mass spectrum. Comprising 23.7% of the particles, this
type is associated with continental dust transport, possibly from mid to low-latitude dust
reaching the Arctic during summer (Willis et al., 2018).
(3) **Potassium-Nickel-Sulfate (K-Ni-SO$_4$):** As seen in Figure 2c, these particles exhibit
prominent $^{39}$K$^+$ and carbonaceous cation signals in the cation mass spectrum, with notable
$^{59}$Ni$^+$ and intense sulfate peaks ($^{97}$HSO$_4^-$) in the anionic mass spectrum. Making up 15.9% of
the particles, their nickel content typically points to petroleum product combustion and other
anthropogenic activities (Shevchenko et al., 2003; Zhan et al., 2014).
(4) **Sulfate (SO$_4$):** Figure 2d illustrates these particles with strong sulfate signal peaks ($^{97}$HSO$_4^-$)
in the anionic mass spectrum and a significant presence of inorganic carbon cations,
constituting 7.6% of the total. Correlations (Figure S2a, r=0.41, p<0.01) observed between the
SO$_4$ and K-Ni-SO$_4$ black carbon particles indicate a common source of anthropogenic origin.
(5) **Potassium-Sulfate (K-SO$_4$):** Shown in Figure 2e, these particles, which account for only
0.6%, are characterized by strong $^{39}$K$^+$ signals in the cationic mass spectrum and sulfate peaks
($^{97}$HSO$_4^-$) alongside inorganic carbon peaks in the anionic mass spectrum. Compared to SO$_4$,
the K-SO$_4$ and K-Ni-SO$_4$ black carbon particles exhibit a stronger correlation (Figure S2b,
r=0.82, p<0.01), suggesting that they also originate from similar human activities.
(6) **Potassium-L-Glucan (K-lev):** Figure 2f depicts these particles with $^{39}$K$^+$ and organic matter
signals in the cationic mass spectrum and characteristic L-glucan peaks (m/z = -45, -59, -71, -
73) in the anionic mass spectrum, indicative of biomass combustion and comprising 0.5% of
the particles (Simoneit et al., 1999).
Previous research has differentiated the sources of Arctic black carbon, identifying fossil fuels as
the primary winter source and biomass burning as the predominant summer source, with an annual
average biomass combustion contribution of 39±10% to atmospheric black carbon (Winiger et al., 2019).
Further studies have shown that biomass combustion contributes over 90% to Arctic ice and snow black
carbon (Hegg et al., 2009). This study's cluster analysis quantitatively estimated that biomass combustion
is the main summer source, accounting for 52.3% of black carbon aerosols, with specific mixed types
including K-CN and K-lev. Anthropogenic and transported sources from mid and low latitudes account
for 24.0% and 23.7%, respectively, with specific mixed types including K-Ni-SO$_4$, SO$_4$, K-SO$_4$, and Ca-



NO$_3$.
**3.2 Spatial distribution of black carbon particles**
As illustrated in Figure S3, the hourly concentration of black carbon particles along the surveyed
route varied substantially, ranging from 0 to 3000 particles per hour. In the initial three segments, the
concentration remained relatively low, with the highest value not surpassing 1500 particles per hour.
Conversely, in Leg IV, the concentration of black carbon particles was markedly higher, reaching a peak
near 3000/h, and Leg V also recorded a significant concentration of 2500/h. The proportion of black
carbon particles varied from 0% to 60% across the route. Unlike the number concentration, the
distribution of the proportion of black carbon particles was more uniform throughout the voyage. Figure
3 displays the hourly number and proportion of black carbon particles by box plots. The mean
concentrations of black carbon particles in Legs I-V were 198/h, 148/h, 255/h, 757/h, and 345/h,
respectively. The corresponding average proportions were 19.8 (±10.7) %, 24.4 (±16.2) %, 26.6
(±13.6) %, 26.1 (±10.9) %, and 26.5 (±10.5) %. Although the average concentration of black carbon
particles varied significantly among the different segments, the proportion of black carbon particles
relative to all particles remained relatively consistent, particularly in the last three segments where the
average proportion showed minimal variation. This consistency suggests that black carbon particles are
prevalent in the Arctic summer ocean boundary layer and constitute a significant and uniformly
distributed component of the atmospheric aerosol. The overall average proportion of black carbon
particles for all segments was 24.7 (±13.6) %.
While the average proportion of black carbon particles remained relatively consistent across
different segments, spatial variations in the proportions of black carbon particle types were observed
(Figure 4), suggesting diverse sources of black carbon aerosols in different regions. In Leg I, the
predominant black carbon types were Ca-NO$_3$, which accounted for about 50% of the particles, followed
by K-CN at approximately 25%, and K-Ni-SO$_4$ and SO$_4$ each at about 10%. K-SO$_4$ and K-lev were nearly
absent. In Leg II, K-CN was the most significant, comprising about 70% of particles, with Ca-NO$_3$ at
around 15%, K-Ni-SO$_4$ and K-lev each at about 5%, SO$_4$ less than 5%, and K-SO$_4$ almost nonexistent.
Leg III saw a dominance of K-CN and Ca-NO$_3$, each making up about 50%, with other types collectively
less than 5%. Leg IV featured K-CN as the most prevalent, at about 60%, with K-Ni-SO$_4$ and Ca-NO$_3$
each contributing around 20%, SO$_4$ at about 5%, and negligible amounts of K-SO$_4$ and K-lev. Leg V had



a similar composition to Leg IV, but with a reduced proportion of Ca-NO$_3$ at about 10% and an increased
proportion of SO$_4$ to about 15%. The distribution of mixed types of black carbon particles helped to
elucidate the source contributions in different regions. In the central Arctic Ocean, the Norwegian Sea-
Iceland, and the North Atlantic Ocean, biomass combustion was the primary source of black carbon,
accounting for more than 50% of particles, particularly in the central Arctic Ocean where it exceeded
70%. In the Chukchi Sea region, terrestrial transport from mid and low latitudes was the dominant source,
also accounting for more than 50% of particles, with biomass combustion and anthropogenic pollution
sources each contributing about 25%. Near Svalbard, biomass combustion sources and terrestrial
transport were the main contributors and were roughly equal in proportion. Previous studies have
indicated that black carbon aerosols from different sources possess varying thicknesses of coatings,
thereby affecting their radiative impacts differently (Matsui et al., 2022). In our study, the sources of
black carbon aerosols in different regions were quantitatively estimated, which is crucial for a more
accurate assessment of the climate effects of Arctic black carbon aerosols.
**3.3 Photochemical processes of Arctic black carbon aerosols**
In urban environments, it has been established that black carbon particles are photoactive, capable
of releasing reactive oxygen species such as excited oxygen molecules ($^1O_2$) and hydroxyl radicals. These
species participate in atmospheric chemical processes, including the catalytic formation of sulfate
(Gehling & Dellinger, 2013; Li et al., 2018; Li et al., 2019; Zhang et al., 2020; Zhang et al., 2021). In the
Arctic Ocean boundary layer, the release of dimethyl sulfur by marine organisms and its subsequent
oxidation to sulfate is known to have significant environmental and climatic impacts (Bates et al., 1987;
Charlson et al., 1987; Rap et al., 2013). However, the potential role of black carbon particles in enhancing
this sulfate formation process remains unclear, a topic we explore in this section. Although black carbon
particles facilitate the catalysis of sulfate formation, they do not appear to significantly promote nitrate
formation (Zhang et al., 2021). Consequently, this leads to a relative enrichment of sulfate compared to
nitrate in particles containing black carbon, increasing the sulfate to nitrate mass concentration ratio
(SNR). In this study, we utilized the ratio of the relative peak areas of sulfate ($^{97}HSO_4^-$) to nitrate ($^{46}NO_2^-$)
to describe SNR, specifically denote the relative enrichment of sulfate in black carbon particles. The
SNR for black carbon particles varied widely, ranging from 0 to 1000, with an average value of 32.25
and a median value of 8.14. This contrasts with measurements from a previous single-particle study of



urban black carbon during summer, which reported SNR values ranging from 0 to 5 and a median value
of about 2 (Zhang et al., 2021). The notably higher SNR in the Arctic summer suggests that emissions of
sulfur-containing gases by marine organisms in the ocean boundary layer may provide a substantial
number of precursors for sulfate generation. Additionally, we calculated the SNR for the most common
sea salt particles in the ocean boundary layer, which ranged from 0 to 6, with an average of 0.34 and a
median of 0.15. This comparison shows that the SNR of black carbon particles is significantly higher
than that of sea salt particles, indicating substantial sulfate enrichment in black carbon particles.
Moreover, we used the cumulative relative peak area values ($R_{BC}$) of inorganic carbonaceous anions ($^{12}C^{+}$,
$^{24}C^{\pm}$, $^{36}C^{\pm}$, $^{48}C^{-}$) in black carbon-containing particles to indicate the content of black carbon components
within the particles. As illustrated in Figure 5, the SNR value increased with the rising content of black
carbon, displaying a moderate correlation ($r = 0.37$, $p < 0.01$). This trend suggests that higher black
carbon content in particles tends to enhance sulfate enrichment over nitrate, indicating that black carbon
components can promote sulfate formation in the Arctic region.
The observed phenomenon of elevated SNR in black carbon particles can be attributed to several
factors: (1) Common Source for Black Carbon and Sulfate Precursors: Black carbon and sulfate, along
with their precursors, possibly share common sources, contributing to the increased SNR in black carbon
particles. Predominantly, black carbon aerosols originate from biomass combustion, which accounts for
over half of these emissions, supplemented by anthropogenic sources and transport from mid to low
latitudes in this study. In the ocean boundary layer, sulfur-containing gases emitted by marine biogenic
sources are primary precursors for sulfate (Becagli et al., 2016; Gondwe et al., 2003; Jarnikova et al.,
2018). However, anthropogenic black carbon aerosols, such as those from ship-based emissions, may
also emit significant amounts of $SO_2$ (Davis et al., 2001; Krause et al., 2021; Winther et al., 2014). After
excluding these anthropogenic sources, the average SNR for black carbon particles from biomass
combustion sources was calculated at 7.85, significantly higher than in sea salt particles. Although higher
$SO_2$ concentrations are noted in biomass combustion air masses, an increase in nitrogen oxides (Leino et
al., 2014) suggests a limited impact on the SNR; (2) Catalysis by Transition Metal Elements: Transition
metals such as iron and vanadium found in black carbon particles can catalyze the formation of sulfate
(Ault et al., 2010; Zhang et al., 2019). These metals typically originate from anthropogenic activities,
including ship-based emissions, and are consequently enriched in anthropogenic black carbon aerosols.



As the concentration of black carbon in the particles increases, so too does the concentration of these
catalytic metals, enhancing their effect on sulfate formation. Despite this, even after accounting for
sources that might skew data, such as those rich in transition metals, black carbon particles still exhibit
a higher enrichment of sulfate compared to sea salt particles. After ruling out these factors, it is evident
that black carbon itself contributes to the enhancement of sulfate formation, aligning with observations
in urban environments (Zhang et al., 2021). To further explore the potential involvement of
photochemical processes in this enhancement, black carbon particles were categorized based on varying
levels of sunlight intensity. The SWGDN served as an indicator of sunlight intensity changes. Black
carbon particles were segmented into five groups based on their SWGDN values, spanning from 0-100,
100-200, 200-300, 300-400, and over 400 W/m^2. As depicted in Figure 6, the SNR values are lower
when the SWGDN value is below 200 W/m^2. However, as the SWGDN value increases, there is a
corresponding rise in SNR values. The average SNR values for black carbon particles across these groups
were 25.2, 21.4, 44.3, 63.1, and 174.4, respectively. These findings underscore a significant impact of
sunlight intensity on the concentration of sulfate in black carbon particles, suggesting that light-driven
photochemical reactions may play a crucial role in this process.

The formation of sulfate and nitrate is intrinsically linked to the presence of oxidizing agents such

as hydroxyl radicals and active halogens, which are often involved in photochemical processes (Chen et
al., 2017; Liu et al., 2021; Shao et al., 2019). Additionally, conditions of strong light are thought to
facilitate the degradation of nitrate, potentially leading to an increased SNR in black carbon particles
(Zhang et al., 2021). These factors may significantly contribute to the observed light intensity-dependent
enrichment of sulfate in black carbon particles. To evaluate this possibility, the SNR values of sea salt
particles under varying lighting conditions, as shown in Figure 6, do not mirror the changes observed in
black carbon particles. The average SNR values for sea salt particles across the five groups—ranging
from low to high sunlight intensities—were 0.33, 0.28, 0.53, 0.52, and 0.33, respectively. Notably, while
there is a slight increase in SNR values for sea salt particles within the 200-400 W/m² range of SWGDN,
the values decrease again when SWGDN exceeds 400 W/m², aligning with the lower values observed
under less intense light conditions. This disparity suggests that while sunlight intensity influences the
photoactivity of black carbon, leading to the release of reactive oxygen species and subsequent sulfate
formation, it does not significantly impact the SNR in sea salt particles. This indicates a unique



photochemical sensitivity in black carbon particles that enhances sulfate formation under higher sunlight
conditions—a process consistent with photochemistry observed in urban environments, rather than being
caused by degradation of nitrate (Zhang et al., 2020; Zhang et al., 2021). Furthermore, the interaction of
sulfate and organic carbon with black carbon aerosols can significantly alter the optical properties of
these particles, thereby increasing their positive radiative forcing (Chung & Seinfeld, 2002). This
photochemical facilitation not only affects the chemical composition of the aerosols but also their
radiative properties, which could have profound implications for the Arctic climate. Such interactions
and their climatic effects underscore the importance of incorporating this photochemistry into future
climate models to accurately predict and mitigate the impacts of black carbon in polar regions.

**4    Summary and Implications**
In this research, we focused on extracting and analyzing black carbon-containing particles from
collected atmospheric samples to understand their mixing states, sources, spatial distribution variations,
and their role in enhancing sulfate formation through photochemical processes. The black carbon
particles were categorized into six distinct groups based on their mixing states: K-CN, Ca-NO$_3$, K-Ni-
SO$_4$, SO$_4$, K-SO$_4$, and K-lev. Our findings indicate that K-CN and K-lev, which are primarily products
of biomass combustion, represent 52.3% of the sampled particles. Ca-NO$_3$, linked to terrestrial sources,
accounted for 23.7% of the particles. The remaining categories—K-Ni-SO$_4$, SO$_4$, and K-SO$_4$—are
predominantly derived from anthropogenic activities, such as emissions from ships, contributing to 24.0%
of the particles. Spatial analysis revealed that the distribution of black carbon-containing particles is
relatively stable across the Arctic regions, suggesting their pervasive presence in the Arctic summer
ocean boundary layer as a significant component of atmospheric aerosols. However, the source
contributions of these particles vary spatially. In the central Arctic Ocean and areas near the Norwegian
Sea-Iceland and the North Atlantic Ocean, more than 50% of the black carbon particles originate from
biomass combustion, with the figure rising above 70% in the central Arctic Ocean. In contrast, in the
Chukchi Sea region, terrestrial transport from middle and low latitudes is the dominant source,
accounting for more than 50% of the black carbon particles, while biomass combustion and
anthropogenic pollution each contribute approximately 25%. Near Svalbard, biomass combustion
sources and terrestrial transport are equally significant contributors to the presence of black carbon



particles. Moreover, our study confirms that black carbon particles in the Arctic Ocean boundary layer
significantly enhance sulfate formation through their involvement in photochemical reactions. This
interaction not only alters the mixing state of the black carbon aerosols but also affects their radiative
properties, potentially influencing the climate. This underscores the importance of understanding the
complex photochemistry of black carbon in Arctic aerosols for future climate modeling and assessment
strategies.

**Acknowledgments**
This work was supported by the National Natural Science Foundation of China (NSFC) (41941014).
We thank China Arctic and Antarctic Administration for fieldwork support.

**Data availability**
Access to the raw data and products is available at DOI:10.5281/zenodo.13883324 or by contacting
either the corresponding author, Zhouqing Xie, at zqxie@ustc.edu.cn, or the first author, Longquan Wang,
at wlq1995@mail.ustc.edu.cn. The SWGDN data featured in this publication can be accessed publicly
through the NASA Goddard Earth Sciences Data and Information Services Center at
https://disc.gsfc.nasa.gov/.

**Declaration of competing interests**
The authors declare that they have no known competing financial interests or personal relationships
that cloud have appeared to influence the work reported in this paper.



**Author contribution:**
Conceptualization: Zhouqing Xie
Data curation: Longquan Wang, Jinpei Yan
Investigation: Longquan Wang, Jinpei Yan
Formal analysis: Longquan Wang, Afeng Chen, Bei Jiang, Fange Yue, Xiawei Yu, Zhouqing Xie
Visualization: Longquan Wang
Funding acquisition: Zhouqing Xie
Supervision: Zhouqing Xie
Writing – original draft: Longquan Wang
Writing – review & editing: Longquan Wang, Zhouqing Xie



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



**Figure Captions**

**Figure 1.** Sampling locations from open water to the high Arctic are indicated by a blue line. The expedition is divided into five segments, each outlined by dashed red boxes: Chukchi Sea section, denoted as Leg I, spans from 22:00 on July 30th, 2017, to 19:00 on August 1st, 2017, covering latitudes from 66.0°N to 74.8°N and longitudes from 169.4°W to 159.2°W. The high Arctic section, marked as Leg II, extends from 20:00 on August 10th to 5:00 on August 12th, 2017, ranging from 83.7°N to 84.6°N and from 132.0°E to 110.4°E. Svalbard Islands section, identified as Leg III, is from 2:00 on August 17th to 11:00 on August 19th, 2017, with latitudes from 82.6°N to 74.3°N and longitudes from 25.4°E to 2.1°E. Norwegian Sea and Iceland section, referred to as Leg IV, occurs from 14:00 on August 23rd to 14:00 on August 25th, 2017, spanning from 67.0°N to 61.2°N and from 2.1°W to 25.8°W. Atlantic Ocean section, labeled as Leg V, spanning from 14:00 on August 25th to 17:00 on August 27th, 2017, covers from 61.1°N to 56.2°N and from 25.9°W to 46.9°W.

**Figure 2.** Spectra of various types of black carbon particles with peaks displaying strong signals highlighted in blue. The X-axis represents the mass-to-charge ratio (m/z), and the Y-axis shows the relative peak area. The cation spectrum is presented in the upper half of each map, while the anion spectrum is depicted in the lower half. (a) Spectra of K-CN black carbon particles; (b) Spectra of Ca-$NO_3$ black carbon particles; (c) Spectra of K-Ni-$SO_4$ black carbon particles; (d) Spectra of $SO_4$ black carbon particles; (e) Spectra of K-$SO_4$ black carbon particles; (f) Spectra of K-lev black carbon particles.

**Figure 3.** Box plots of hourly counts (in red) and fractions (in blue) of black carbon particles from Leg I to Leg V. The tops and bottoms of the boxes indicate the upper and lower quartile values, respectively. Horizontal lines within the boxes denote median values. Hollow squares and dotted black lines represent mean values. Vertical whiskers extend to scattered values above and below the boxes, with caps on the whiskers indicating maximum and minimum values. Cross symbols mark the 99th and 1st percentile values, respectively.

**Figure 4.** Fraction of each type of black carbon particles from Leg I to Leg V. Different colors represent various types of black carbon particles: blue for K-CN, green for Ca-$NO_3$, orange for K-Ni-$SO_4$, magenta for $SO_4$, cyan for K-$SO_4$, and red for K-lev.

**Figure 5.** Correlation of SNR and $R_{BC}$ in black carbon particles. A red dashed line indicates the linear fit curve, with the correlation coefficient (r) and significance level (p) provided.





**Figure 6**. Box plots of SNR for sea salt particles (in red) and black carbon particles (in blue), grouped

by SWGDN. The upper and lower parts of the boxes indicate the quartile values, while the horizontal

lines within the boxes denote the median values. Hollow squares and black dotted lines show the mean

values. Vertical whiskers extend to capture scattered values above and below the boxes, with caps on the

whiskers indicating the maximum and minimum values. Cross symbols mark the 99th and 1st percentile

values, respectively.



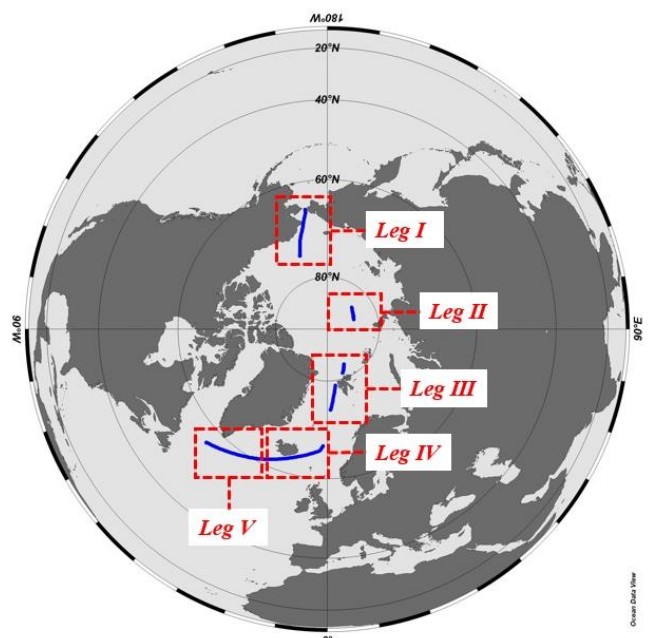

**Figure 1.** Sampling locations from open water to the high Arctic are indicated by a blue line. The expedition is divided into five segments, each outlined by dashed red boxes: Chukchi Sea section, denoted as Leg I, spans from 22:00 on July 30th, 2017, to 19:00 on August 1st, 2017, covering latitudes from 66.0°N to 74.8°N and longitudes from 169.4°W to 159.2°W. The high Arctic section, marked as Leg II, extends from 20:00 on August 10th to 5:00 on August 12th, 2017, ranging from 83.7°N to 84.6°N and from 132.0°E to 110.4°E. Svalbard Islands section, identified as Leg III, is from 2:00 on August 17th to 11:00 on August 19th, 2017, with latitudes from 82.6°N to 74.3°N and longitudes from 25.4°E to 2.1°E. Norwegian Sea and Iceland section, referred to as Leg IV, occurs from 14:00 on August 23rd to 14:00 on August 25th, 2017, spanning from 67.0°N to 61.2°N and from 2.1°W to 25.8°W. Atlantic Ocean section, labeled as Leg V, spanning from 14:00 on August 25th to 17:00 on August 27th, 2017, covers from 61.1°N to 56.2°N and from 25.9°W to 46.9°W.

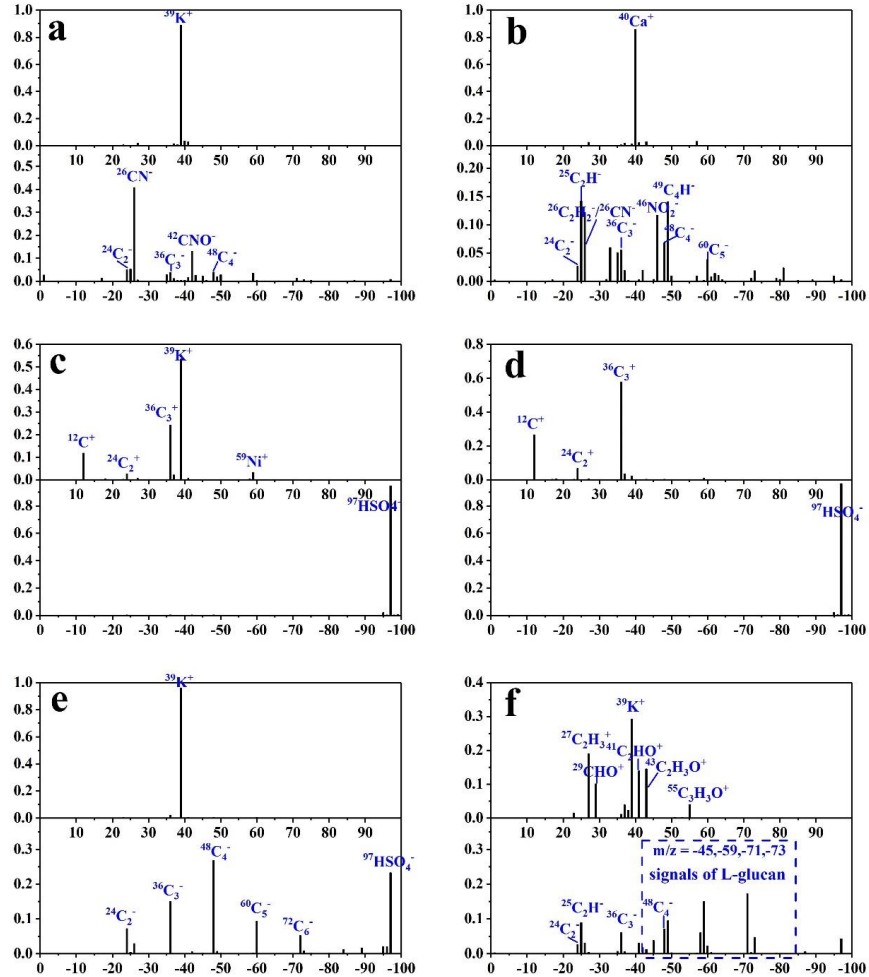

**Figure 2.** Spectra of various types of black carbon particles with peaks displaying strong signals highlighted in blue. The X-axis represents the mass-to-charge ratio (m/z), and the Y-axis shows the relative peak area. The cation spectrum is presented in the upper half of each map, while the anion spectrum is depicted in the lower half. (a) Spectra of K-CN black carbon particles; (b) Spectra of Ca-NO₃ black carbon particles; (c) Spectra of K-Ni-SO₄ black carbon particles; (d) Spectra of SO₄ black carbon particles; (e) Spectra of K-SO₄ black carbon particles; (f) Spectra of K-lev black carbon particles.





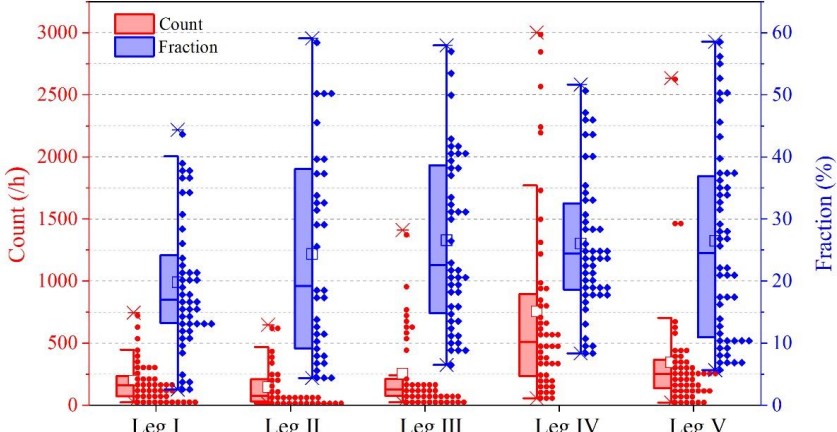

653

**Figure 3.** Box plots of hourly counts (in red) and fractions (in blue) of black carbon particles from Leg

I to Leg V. The tops and bottoms of the boxes indicate the upper and lower quartile values, respectively.

Horizontal lines within the boxes denote median values. Hollow squares and dotted black lines represent

mean values. Vertical whiskers extend to scattered values above and below the boxes, with caps on the

whiskers indicating maximum and minimum values. Cross symbols mark the 99th and 1st percentile

values, respectively.

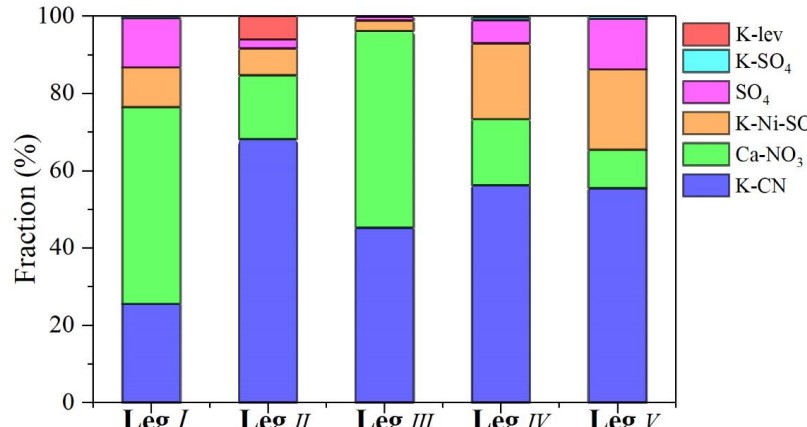

660

**Figure 4.** Fraction of each type of black carbon particles from Leg I to Leg V. Different colors represent

various types of black carbon particles: blue for K-CN, green for $Ca-NO_3$, orange for $K-Ni-SO_4$, magenta

for $SO_4$, cyan for $K-SO_4$, and red for K-lev.



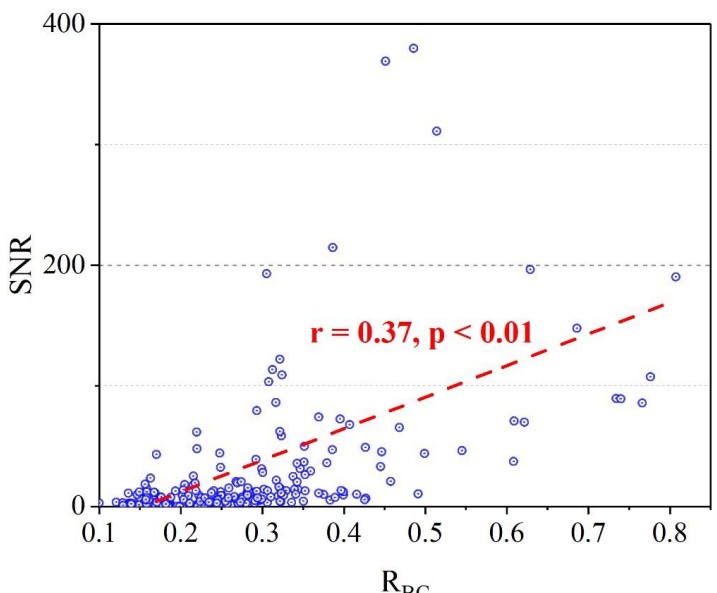

664

**Figure 5.** Correlation of SNR and $R_{BC}$ in black carbon particles. A red dashed line indicates the linear fit

curve, with the correlation coefficient (r) and significance level (p) provided.

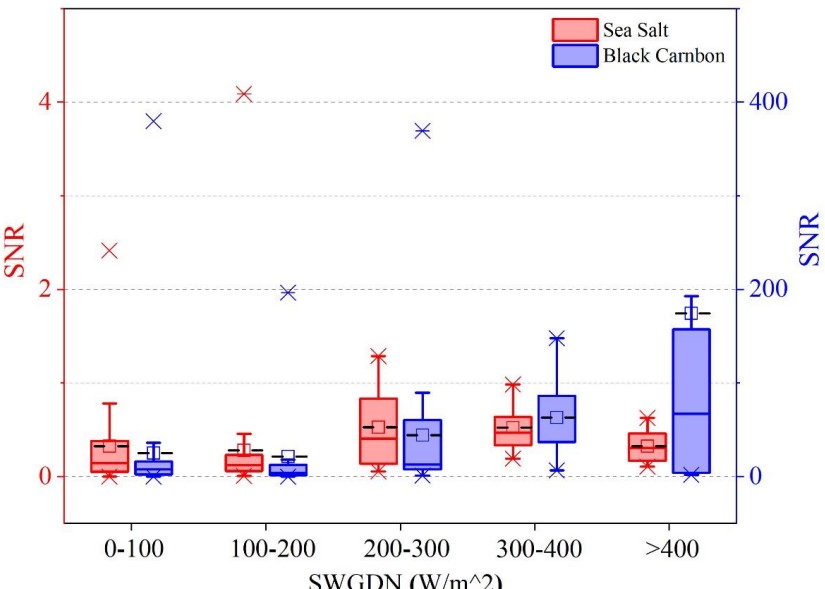

667

**Figure 6**. Box plots of SNR for sea salt particles (in red) and black carbon particles (in blue), grouped

by SWGDN. The upper and lower parts of the boxes indicate the quartile values, while the horizontal

lines within the boxes denote the median values. Hollow squares and black dotted lines show the mean

values. Vertical whiskers extend to capture scattered values above and below the boxes, with caps on the

whiskers indicating the maximum and minimum values. Cross symbols mark the 99th and 1st percentile

values, respectively.