# Peer review of "Mixing state, spatial distribution, sources and"

_EGUsphere, 2024_

## Author Comment (AC2)

AUTHORS' RESPONSE TO COMMENTS FROM EDITORS AND REVIEWERS

We thank the reviews and the editor for their detailed reviews. Please see our responses (in regular type) below the reviewer (in bold type).

**Review (Major Comment):**
**Q: Summary of SPAMS measurements: The SPAMS have been deployed many times on R/V Xuelong for the Chinese Arctic Research Expedition. There are a handful of papers about the SPAMS measurement, not limited to Wang et al. (2022) and Su et al. (2024). I am unsure if black carbon measurement using SPAMS has been reported before. I suggest the author first summarize what has been done using SPAMS on R/V Xuelong in a table in the supplement and then outline the novelty of the work in the Introduction. This will greatly help the readers understand the progress of similar work using SPAMS.**
Re: We added a comprehensive summary table (Table S2) detailing the previous applications of SPAMS aboard the R/V Xuelong. Additionally, we have expanded the text in lines 175-180 to highlight the innovative aspects of this study compared to previous research.

**Q: Mass spectra of particles: My understanding of the mass spectra in Figure 2 is that each of them represented the average mass spectra of particles clustered by ART-2a. If that is the case, error bars should be included to indicate the ranges of the ions for individual ions shown in the mass spectra. When grouping the ions, did the authors exclude what has been reported in Wang et al. (2022) and Su et al. (2024)? However, it appears to me that this is not the case. By visually comparing the mass spectra from this study with those in the Wang et al. (2022) and Su et al. (2024), I suspect there are some overlaps between the presented mass spectra:**
**Fig 2b in Wang et al. (2022) looks like the mix of Fig 2a, f in this study;**
**Fig 2a in Wang et al. (2022) has some degrees of similarities to Fig 2b in this study;**
**Fig 2g OC-S in Su et al. (2024) is like the Fig 2c in this study;**
**Fig 2d OC-K in Su et al. (2024) is like the Fig 2f in this study**
**More details should be provided about how black-carbon-containing particles were defined. In addition, the mass spectra in this study should be quantitatively compared with those reported in Wang et al. (2022) and Su et al. (2024), with the aid of contrast angle.**
Re: ART-2a is an unsupervised, real-time clustering algorithm that categorizes input data by dynamically adjusting class centers, known as prototype vectors. Its operation is based on competitive learning and a vigilance parameter, distinguishing it from statistical-based clustering methods like K-means, which typically compute within-cluster variance or confidence intervals. Unlike these methods, ART-2a does not automatically generate error bars because its class centers are deterministic outcomes rather than probabilistic distributions. Consequently, the average spectra derived from ART-2a clustering do not include error bars.

In this study, we configured ART-2a with a vigilance factor of 0.6, a learning rate of 0.05, and a maximum of 20 iterations, following previous research guidelines to capture over 95% of total particles for classification purposes. Prior to clustering, we filtered the particles under specific conditions, selecting only those exhibiting at least three concurrent cationic or anionic carbon peaks. From approximately 290,000 ionized particles, we identified about 80,000 that we believe to be black carbon particles or particles containing black carbon components. In contrast, Wang isolated

393 iodine-containing particles for analysis based on the presence of m/z = -127, while Su performed a broader clustering analysis on all ionized particles, categorizing approximately 38.6% as inorganic and 61.4% as organic. Our selective process ensures that we do not inadvertently exclude any target particles when employing the ART-2a algorithm, maintaining objectivity in our analysis. Given that the particles we selected are a subset of those analyzed by Su, some spectral similarities are expected. However, similarities in spectra do not necessarily imply identical chemical properties or processes. For instance, Figure 2b in Wang et al. (2022) shows spectral similarities to those in our study but also includes iodine signals, suggesting distinct chemical interactions. Therefore, a direct visual comparison of the clustered spectra from our study with those from Wang and Su may not yield meaningful insights.

**Q: What is the size distribution of the detected particles? What is the size-resolved distribution of the observed aerosol types?**

Re: The size distributions of ionized particles and total black carbon particles are presented in Figure S2 and discussed in lines 205-207 of the text. Additionally, the size-resolved distributions of the observed aerosol types are depicted in Figure 2(g) and detailed in lines 286-293.

**Q: Section 3.2: The current analysis lacks details and is descriptive. The message presented in Figs 3 and 4 is too generalized. There is no information about the backward air mass trajectories. It is important to know the time spent on open water, land, and sea ice as well as the time spent within the mixing layers. Readers would be interested in the source region of individual aerosol types in each leg. Such analysis can be carried out using potential source contribution function (PSCF) or concentration-weighted trajectory (CWT). In addition, the information about the size-resolved aerosol types should be included, similar to those in Su et al. (2024)**

Re: we added backward air mass trajectories and PSCF analysis in Figure 3(a)-(e) and Figure 3(h) and discussed in section 3.2.

**Q: Section 3.3: The author speculates the high SNR was due to the sulfur-containing gases from marine organisms. Did the MSA signal coincide with the high SNR values?**

Re: Due to the susceptibility of the signal peaks of MSA ($^{95}CH_3SO_3^-$) in single-particle mass spectrometry to interference from $^{95}PO_4^-$ and $^{95}NaCl_2^-$, it is not possible to directly ascertain the concentration of MSA from these measurements. Consequently, our suggestion that MSA concentrations might be higher is speculative. However, this hypothesis appears plausible as evidenced by the higher SNR observed in open sea areas compared to sea ice areas, as illustrated in Figure 4a.

**Q: Lines 330 – 332: How did you exclude the anthropogenic sources? What were the major aerosol types left for the analysis?**

**Re:** In the previous version of our manuscript, we identified the peak at m/z = 59 as indicative of nickel ($^{58}Ni^+$) and speculated that particles containing this signal might originate from ship-based fossil fuels. However, following insights from another reviewer, we now understand that this signal peak should be attributed to $^{59}C_3H_9N^+$, not Ni. We have corrected this error in the current version and have revisited and revised the discussion accordingly in lines 404-410.

**Minor Comment:**

**Q: Line 155: What was the in-line RH after the Nafion dryer?**

Re: The relative humidity (RH) after passing through the Nafion dryer is maintained below 40%. We have included this relevant information in the text at line 160.

**Q: Line 155: Section 2.2: How many particles have been sampled? I found two different numbers in Wang et al. (2022) and Su et al. (2024), which used the same data set as this study, i.e., "More than 2,000,000 particles" vs. "over 2.25 million particles".**

Re: The precise values are 2398509 particles were sampled and sized while 286079 particles were ionized with positive and negative mass spectrometry available. We modified our description in lines 189-192.

**Q: Section 2.2: How was the influence of ship exhaust from the research vessel removed? I am concerned if the measurement has been affected by the ship exhaust from the research vessel, which can emit black carbon. I also don't see any details about detecting ship exhaust in Wang et al. (2022) and Su et al. (2024).**

Re: During the research vessel's voyage, we implemented several measures to minimize ship-based pollution. For example, we positioned the sampling ports on the top deck and restricted sampling activities to periods when the vessel was at sea. More critically, we assessed the impact of ship-based emissions by analyzing the spectral data. Typically, the presence of signal peaks at m/z $^{51}V^+$ and $^{67}VO^+$ in the spectra would indicate the influence of the vessel's heavy oil combustion emissions. However, these specific signal peaks were not detected in the black carbon particles we identified, suggesting that the influence of ship-based emissions was minimal. Additionally, it is possible that we inadvertently excluded a small fraction of contaminated particles during the ART-2a cluster analysis, as we only analyzed the top 95% of particles following clustering.

**Q: Figure 2: I will suggest the authors use different colors to differentiate different groups of ions (e.g., C-only ions, oxygenated C ions, metal ions)**

Re: We have updated the graph in Figure 2 as suggested. In the revised figure, carbon ion clusters ($^{12}C^+$, $^{24}C_2^{\pm}$, $^{36}C_3^{\pm}$, $^{48}C_4^-$, $^{60}C_5^-$, $^{72}C_6^-$) are now marked in black. Metal ions are indicated in red, organic ions in orange, inorganic anions in blue, and other ions are depicted in grey. This color-coding enhances the clarity and readability of the data presented in the figure.

**Q: Section 3.3: Could you include a histogram illustrating the SNR?**

Re: we added a histogram illustrating SNR in Figure 4a.

**Q: Could you please include the mass spectra of sea salt particles in the supplement?**

Re: we added mass spectra of sea salt in Figure S1b.

**Q: What should be the SNR we should expect for aerosol types associated with biomass combustion air masses?**

**Re:** We assess whether sulfate is relatively enriched in black carbon particles by comparing it to the SNR in sea salt particles. Clearly, we do not have an expected value for this comparison, as there are no existing research studies available that provide a reference.

**Q: Figure 5: Please color the data point left after excluding the anthropogenic sources (line 331). In addition, the correlation analysis should be carried out only with the data points after excluding the influence of anthropogenic sources.**

Re:In light of the reasons outlined above (referenced in the last question under Major Comments), we have revised the discussion section in lines 404-410.

**Q: Figure 6: For certain groups, the hollow squares are outside the interquartile ranges. What does it mean? This may happen when there are a few very large values in a small sample size. Could you please clean the data properly before making the analysis? In addition, it will be better to use violin plots to visualize the distribution of the data and list the number of particles for the two particle types in each SWGDN group.**

Re:Indeed, we identified some extremely high values in our data. In Figure 4c, after removing these outliers, we observed an improved linear relationship between SNR and $R_{BC}$, confirming that the observed linearity was not skewed by a few anomalous data points. Furthermore, in the creation of Figure 5 (previously Figure 6), we not only removed outliers but also incorporated violin plots to provide a more detailed representation of the data distribution.

**Technical Comment:**
**Q: Line 109: higher and longer compared to what?**

Re: The sentence was revised to "Moreover, Asian anthropogenic black carbon typically travels at higher altitudes and over longer distances than those from Europe, Siberia, and North America, resulting in a longer residence time and more thorough aging process (Matsui et al., 2022)" in lines 110-113.